# The Gaze Fixation Assessed by Microperimetry: A Useful Tool for the Monitoring of the Cognitive Function in Patients with Type 2 Diabetes

**DOI:** 10.3390/jpm11080698

**Published:** 2021-07-22

**Authors:** Ángel Michael Ortiz-Zúñiga, Olga Simó-Servat, Alba Rojano-Toimil, Julia Vázquez-de Sebastian, Carmina Castellano-Tejedor, Cristina Hernández, Rafael Simó, Andreea Ciudin

**Affiliations:** 1Vall d’Hebron Research Institute, Universitat Autònoma de Barcelona (VHIR-UAB), 08035 Barcelona, Spain; a.ortiz@vhebron.net (Á.M.O.-Z.); olga.simo@vhir.org (O.S.-S.); julia.vazquez@vhir.org (J.V.-d.S.); cristina.hernandez@vhir.org (C.H.); 2Endocrinology and Nutrition Department, Hospital Universitari Vall Hebron, 08035 Barcelona, Spain; arojano@vhebron.net; 3CIBER of Diabetes and Metabolic Disease, Instituto de Salud Carlos III, 28029 Madrid, Spain; 4RE-FiT Barcelona Research Group, Vall d’Hebrón Institute of Research & Parc Sanitari Pere Virgili, 08023 Barcelona, Spain; ccastellano@perevirgili.com; 5GIES Research Group, Basic Psychology Department, Autonomous University of Barcelona, Bellaterra, 08192 Barcelona, Spain

**Keywords:** type 2 diabetes, retinal microperimetry, cognitive impairment

## Abstract

Current guidelines recommend annual screening for cognitive impairment in patients > 65 years with type 2 diabetes (T2D). The most used tool is the mini-mental state evaluation (MMSE). Retinal microperimetry is useful for detecting cognitive impairment in these patients, but there is no information regarding its usefulness as a monitoring tool. We aimed to explore the role of retinal microperimetry in the annual follow-up of the cognitive function of patients with T2D older than 65 years. Materials and Methods: Prospective observational study, comprising patients > 65 years with T2D, attended at our center between March–October 2019. A complete neuropsychological evaluation assessed the baseline cognitive status (mild cognitive impairment, MCI, or normal, NC). Retinal microperimetry (sensitivity, gaze fixation) and MMSE were performed at baseline and after 12 months. Results: Fifty-nine patients with MCI and 22 NC were identified. A significant decline in the MMSE score was observed after 12 months in the MCI group (25.74 ± 0.9 vs. 24.71 ± 1.4; *p* = 0.001). While no significant changes in retinal sensitivity were seen, all gaze-fixation parameters worsened at 12 months and significantly correlated with a decrease in the MMSE scores. Conclusion: Retinal microperimetry is useful for the monitoring of cognitive decline in patients > 65 years with T2D. Gaze fixation seems a more sensitive parameter for follow-up after 12 months than retinal sensitivity.

## 1. Introduction

Current epidemiological data show that type 2 diabetes (T2D) increases the risk of developing Alzheimer’s disease (AD) by two to three times in comparison with the non-diabetic population matched by age and other risk factors [1].

Mild cognitive impairment (MCI) represents a deterioration in cognition superior to that observed with normal cognitive decline associated with age. This decline is not severe enough to cause impaired daily function. [2] According to data, about 33% of women and 20% of men over 65 years of age with MCI could develop dementia [3]. Recent data have shown T2D to be an independent risk factor, alongside the presence of εthe APOEε4 allele, for conversion to dementia in patients with MCI [4,5]. The presence of MCI can lead to mismanagement of the specific treatment for T2D. This can cause episodes of severe hypoglycemia and increasing cardiovascular mortality, and hypoglycemia itself worsens the patient’s cognitive state, all of which translates into an increase in healthcare costs directly or indirectly [6]. For this reason, it is important to diagnose cognitive impairment in the early stages in patients with T2D. This makes it possible to implement a patient-centered treatment based on its simplicity and the prioritizing of antidiabetic treatments with a low capacity to provoke hypoglycemia.

The American Diabetes Association (ADA) clinical guidelines recommend screening for the early detection of cognitive impairment for adults ≥ 65 years of age with T2D at the initial visit and annually, as appropriate [7]. For this purpose, screening tools such as the mini-mental state examination (MMSE) test and the Montreal cognitive assessment (MoCA) are recommended. Nevertheless, once the screening is performed, diagnosis of MCI is based on a complex neuropsychological test [2] that makes infeasible their incorporation into current standards of care for the whole population with T2D. At present, there are no reported reliable examinations or phenotypic indicators to identify patients with T2D with MCI.

Retinal microperimetry is a simple, rapid, and non-invasive test that measures retinal sensitivity in terms of the minimum light intensity that patients can perceive when spots of light stimulate specific areas of the retina and also evaluates gaze-fixation stability. [8]. We have previously shown that retinal microperimetry is a useful tool for the identification of MCI in patients with T2D > 65 years old [4]. Retinal sensitivity significantly correlated with brain imaging, such as magnetic resonance imaging (MRI) and positron emission tomography (PET) and identified patients with MCI and dementia, as confirmed by complete neuropsychological battery. Furthermore, by adding the gaze-fixation parameters to the retinal sensitivity, the probability of identifying cognitive impairment significantly increased in an independent manner, suggesting that retinal sensitivity and gaze fixation explore different neuronal circuits and are complementary [5]. Nevertheless, at present, there are no data regarding the usefulness of retinal microperimetry as a monitoring tool for assessing cognitive decline.

On these bases, the main objectives of the present study were: (1) to validate our previous results showing that retinal microperimetry is a reliable screening tool for MCI in patients with T2D patients and (2) to explore the usefulness of retinal microperimetry as a tool for the monitoring of the cognitive function at 12-month follow-up.

## 2. Materials and Methods

This was a prospective, observational pilot study which comprised 100 consecutive patients > 65 years with T2D and without known cognitive impairment attending the outpatient clinic of our center between March–October 2019. The study was conducted according to the tenets of the Helsinki Declaration. The Ethic Committee of the Vall d’Hebron University Hospital approved all procedures (PR(AG)28/2017), and it was conducted following the Strengthening of the Reporting of Observational Studies in Epidemiology guidelines.

The inclusion criteria were: (a) age between 65–85 years old; (b) T2D with a duration > 5 years; (c) functional literacy, without severe auditory or visual abnormalities; (d) without or with mild to moderate diabetic retinopathy; (e) the absence of other eye degenerative diseases apart from diabetes (e.g., glaucoma); and (f) written informed consent. The exclusion criteria were: (a) other types of diabetes; (b) a previous diagnosis of AD or other types of dementia following the diagnostic criteria of the National Institute on Aging and Alzheimer’s Association (NIA-AA) [9]; (c) a history of stroke; (d) a history of unstable neurological or psychiatric conditions that may affect cognition, including the presence of depression; (e) severe metabolic or systemic disease, such as unstable acute cardiovascular disease, renal failure with glomerular filtration rate (GFR) < 30 mL/min/m^2^, decompensated liver cirrhosis or liver failure, or untreated hypothyroidism or vitamin B12 deficiency; (f) drugs affecting cognitive status, for example, antipsychotics, opioids, or long half-life benzodiazepines; (g) uncorrected severe sensory deficits (blindness, deafness); (h) severe diabetic retinopathy (photocoagulation, etc.); (i) or other neurodegenerative diseases (e.g., Parkinson’s).

All the evaluated patients underwent a complete medical history, physical examination, and biochemical analysis (including HbA1c, lipidic profile, the microalbumin/creatinine excretion ratio). A complete neuropsychological evaluation was performed by a trained neuropsychologist using the repeatable battery for the assessment of neuropsychological status (RBANS). RBANS is an instrument that quickly allows a clinician to assess a variety of cognitive domains and to classify patients as normo-cognitive (NC), MCI, or with dementia [10]. The patients classified as MCI were offered the opportunity to be included in the study. Those that gave their informed consent and were recruited for follow-up underwent retinal microperimetry and the mini-mental state evaluation (MMSE), validated in Spanish, [11] at baseline and 12-month follow-up. The cut-off for a normal cognitive function score of MMSE is ≥26 (range 0–30). 

Retinal sensitivity and gaze-fixation parameters were evaluated by fundus-driven microperimetry (MAIA 3rd generation) after a previous pupillary dilation of a minimum of 4 mm. The standard MAIA test covers a 10-diameter area with 37 measurement points, and a red 1 radius circle was used as the fixation target. A four-level fixed strategy was used: Goldmann III size stimulus, background luminance of 4 as band maximum luminance of 1000 asb, with a 25-dB dynamic range. The microperimeter automatically compensates for eye movements during examination via a software module that tracks them. For the evaluation of fixation, the MAIA uses high-speed eye trackers (25 Hz) to record the fixation pattern and to control fixation losses. To calculate the fixation indexes P1 and P2, the MAIA uses all fixation positions during the test, representing the percentage of fixation points within a circle of 2 and 4 degrees in diameter, respectively. Using the bivariate contour ellipse area (BCEA), the MAIA provides a more accurate fixation pattern station. The BCEA is calculated as the bivariate ellipse (measured in squared degrees “º^2^”) that covers the fixing positions and takes into account 1 standard deviation (BCEA63%) or 2 standard deviations (BCEA95%). Through the major and minor axes, the area of this ellipse (2 orthogonal diameters) is calculated, describing the extension of the fixation points (horizontal and vertical diameters). For the analysis, we used data from the first explored eye (usually the right eye). The main variables included in the study were the MMSE score and the specific domains disclosed of the MMSE scores: temporal orientation, spatial orientation, immediate recall, attention, delayed recall, and language.

This was a pilot study. At present, there are no similar studies; therefore, the sample size was not calculated. 

Statistical analysis: The categorical variables were expressed as a percentage. For quantitative variables that follow a normal distribution, they were expressed as means and standard deviation; for those that did not, they were expressed as median and range. The chi-square test was used to evaluate the differences between groups for qualitative variables that follow a normal distribution. On the other hand, for the quantitative variables that follow a normal distribution, the ANOVA test was used. The non-parametric Kruskal-Wallis test was used to evaluate the differences between groups of variables that do not follow a normal distribution. For the 12-month follow-up on the quantitative variables, we used the Student’s *t*-test to compare means for paired data.

To evaluate the correlation between MMSSE and MAIA values, the Pearson correlations were performed. Significance was accepted at *p* < 0.05. The Bonferroni correction was used for multiple comparisons. Statistical analyses were performed with the STATA 15 statistical package.

## 3. Results

At baseline, from the total of 100 patients with T2D that were evaluated using RBANS, 31 were catalogued as NC, 64 as MCI, and 5 as having dementia. Of these patients, 59 with MCI and 22 with NC agreed to be enrolled in a 12-month follow-up observational study. The baseline characteristics are shown in Table 1.

Both groups were similar in terms of the BMI, cardiovascular risk factors, diabetes duration, insulin use, the degree of metabolic control, and the presence of micro- and macroangiopathic complications. The NC group was significantly younger than the MCI group, but the logistic regression model (including retinal sensitivity, gaze-fixation parameters, gender, age, and T2D duration) showed no significant influence of age on the cognitive impairment status in our study (Table 2). 

Regarding microperimetry, significant differences were seen between the two groups at baseline in all the parameters that were evaluated, as reflected by Table 3. 

At baseline, both retinal sensitivity and gaze fixation were independently correlated with the diagnosis of MCI (AUROC 0.7703, *p* < 0.01, and AUROC 0.765, *p* < 0.01, respectively). When the two variables were combined, the predictive value for MCI detection increased (AUROC 0.826, *p* = 0.031), confirming in this cohort our previous results [5].

As expected, we found differences in the MMSE score at baseline between the two groups (28.86 ± 0.8 for NC and 25.74 ± 0.9 for MCI; *p* < 0.001). At 12-month follow-up, a significant decrease in MMSE was seen in the group of MCI patients (25.74 ± 0.9 at baseline vs. 24.97 ± 1.4 after 12 months, *p* = 0.01), while the MMSE score remained unchanged in the NC group (28.86 ± 0.8 vs. 28.32 ± 0.8, *p* = n.s.). The analysis of the paired data of the microperimetry parameters showed that retinal sensitivity did not change after 12 months in the patients included in the study. By contrast, significant worsening was found in all the parameters related to gaze fixation (Table 4). The registers of gaze fixation in a representative patient in whom a worsening of MMSE is displayed in Figure 1.

We found significant correlations between MMSE and microperimetry parameters at baseline both in retinal sensitivity (r = 0.362, *p* = 0.009) and gaze fixation (BCEA63 r = −0.267, *p* = 0.018, BCEA95 r = −0.289, *p* = 0.0133). 

After 12 months follow-up, no significant correlation was seen between MMSE and retinal sensitivity (r = 0.06, *p =* 0.615), while we found a significant negative relationship with gaze-fixation parameters (BCEA95 r = −0.65, *p* = 0.001, BCEA63 r = −0.49, *p =* 0.001). Furthermore, MMSE at 12-month follow-up showed a significant correlation with gaze-fixation parameters when disclosed by the specific domains. Attention (P1: r = −0.306, *p* = 0.0198, P2: r = −0.322, *p* = 0.0128, BCEA63: r = 0.274, *p* = 0.037, BCEA95: r = 0.291, *p* = 0.02) and delayed recall (P1: r = 0.304, *p* = 0.0190, P2: r = 0.313, *p* = 0.0155, BCEA63: r = −0.376, *p* = 0.003, BCEA95: r = −0.33, *p* = 0.0113) significantly correlated with all gaze-fixation parameters. Additionally, the change in MMSE after 12 months (ΔMMSE) showed a significant indirect relationship with the changes in gaze fixation during follow-up ΔMMSE- ΔBCEA63 (r = −0.3217, *p* = 0.0092); ΔMMSE- ΔBCEA95 (r = −0.3396, *p* = 0.015). No other significant correlation was seen in this analysis.

## 4. Discussion

As far as we know, this is the first prospective study to evaluate the usefulness of retinal microperimetry not only as a tool for screening but also for the monitoring of cognitive function in patients with T2D > 65 years old. 

The baseline evaluation confirmed and extended our previous results that retinal sensitivity and gaze-fixation parameters measured by microperimetry are useful for the detection of cognitive impairment in patients with T2D > 65 years [4,5]. 

For inclusion in the study, we used a complete neuropsychological evaluation performed by a trained neuropsychologist using the repeatable battery for the assessment of neuropsychological status (RBANS). RBANS takes about 45 min to assess a variety of cognitive domains and to classify patients as normo-cognitive (NC), MCI, or having dementia [10]. For the prospective study we chose MMSE, which is one of the tests recommended in the ADA guidelines [7] to be used in clinical practice in patients with T2D > 65 years both for screening and for annual monitoring. ADA guidelines also recommend for this purpose the Montreal cognitive assessment (MoCA) test [12]. However, MoCA needs previous training and certification, making MMSE the most accessible and widely used score for the evaluation of cognitive function in daily clinical practice.

In the present study, we found significant correlations between MMSE and both retinal sensitivity and gaze-fixation parameters at baseline. However, only gaze-fixation parameters (P1, P2, BCEA63, BCEA95) showed significant worsening at the end of 12 months of follow-up and were correlated to the global score of MMSE. Furthermore, the worsening of MMSE was parallel to the worsening in gaze fixation, suggesting that retinal microperimetry, specifically gaze fixation, can be a reliable and more subtle tool for the monitoring of the cognitive function in patients with type 2 diabetes. Particularly, such cognitive domains as attention and delayed recall significantly correlated with all gaze-fixation parameters during follow-up.

These results suggest that since retinal sensitivity is a reliable screening tool for diagnosis, the evaluation of gaze fixation could represent a better biomarker for annual follow-up. As previously explained [5], this finding could be attributed to the fact that the areas of the brain involved in gaze fixation are different from those of retinal sensitivity. The sensitivity of the retina is based on the retina and the neural pathway of vision, which includes the geniculate body. Our group showed previously that retinal sensitivity correlated with grey matter volumes in brain MRI in T2D patients with MCI and AD but not white matter [4], suggesting that retinal sensitivity is a marker of neurodegeneration. One can hypothesize that neurodegeneration does not significantly change over 12 months.

On the other hand, gaze fixation depends on the complex white matter network. The brain structures that play an essential role in gaze fixation are the superior colliculus and parietal and frontal cortex. [13]. It should be noted that the retina has several types of ganglion cells, most of them directly connected to the geniculate body of the visual pathway, but others are connected directly to the superior colliculus [14]. The later ones respond best to small stimuli and abrupt changes in light intensity. The main principle of retinal microperimetry is projecting light stimuli of different intensities over a short period of time, suggesting that first, it might directly stimulate the superior colliculus, possibly explaining the early impairment of fixation in comparison with sensitivity. 

Additionally, attention and delayed recall, which were significantly correlated in our study with gaze-fixation parameters, depend mainly on the default brain network and the white-matter connections between the dorsolateral and prefrontal cortex and inferior and superior parietal lobules [15,16]. In support of this data, our group previously showed that gaze fixation was impaired in young patients with obesity and significantly correlated with cognitive function, in particular with attention span, while retinal sensitivity was not altered and did not correlate with the neurocognitive scores [13]. These results suggest that alterations in white matter connectivity are more subtle and occur earlier than neurodegeneration in the visual pathway, thus explaining why the alterations in both gaze fixation and cognitive domains (attention and delayed recall) could already be seen at 12 month follow-up, while retinal sensitivity remained unchanged.

Our study has certain limitations, such as the fact that RBANS was not repeated at 12 months in order to assure an accurate diagnosis of the evolution of cognitive function. Nevertheless, it should be noted that we aimed to explore the utility of retinal microperimetry as a simple tool for the annual monitoring of the cognitive function in patients with T2D > 65 years, replicating the recommendations of the actual ADA guidelines and daily clinical practice. At present, it is impossible to implement a complete battery of neuropsychological tests for all T2D patients > 65 years, and even MMSE would be time consuming if we used it for the evaluation of all patients with T2D > 65 years. Another important limitation of the neuropsychological tests, including MMSE, is that the patient´s performance depends on both emotional mood and educational level. The associated anxiety-depression that happen in a significant proportion of ageing patients with T2D is not generally considered [17]. Is should be noted that the prevalence of depression is two-fold higher in patients with T2D compared with the general population worldwide [18], and it has recently been reported as 27.5% among T2D patients in the Mediterranean population [19]. Retinal microperimetry, used in daily clinical practice as part of the ophthalmological evaluation, is a simple, objective, and rapid test that can be largely used for the monitoring of cognitive performance regardless of psychological status or educational level.

In this study, we confirmed our previous results that retinal microperimetry is a useful tool for detecting MCI in patients with T2D > 65 years. Additionally, our results suggest that gaze-fixation parameters could represent a better marker for monitoring the cognitive function annually, as ADA guidelines recommend [7]. From a practical point of view, we propose the implementation of automatic software in our clinics in order to provide reliable data on those patients with T2D > 65 years that present significant changes in gaze fixation annually, as possible patients at a higher risk of the progression of cognitive impairment and on whom diabetes care providers should focus both for an accurate diagnosis and for their subsequent management. These preliminary data have the potential to change the current guidelines regarding clinical practice in patients with T2D. Nevertheless, further studies are needed in order to validate our results in larger cohorts of patients and to provide a reliable cut-off for the detection of the progression of cognitive decline in T2D patients.

## Figures and Tables

**Figure 1 jpm-11-00698-f001:**
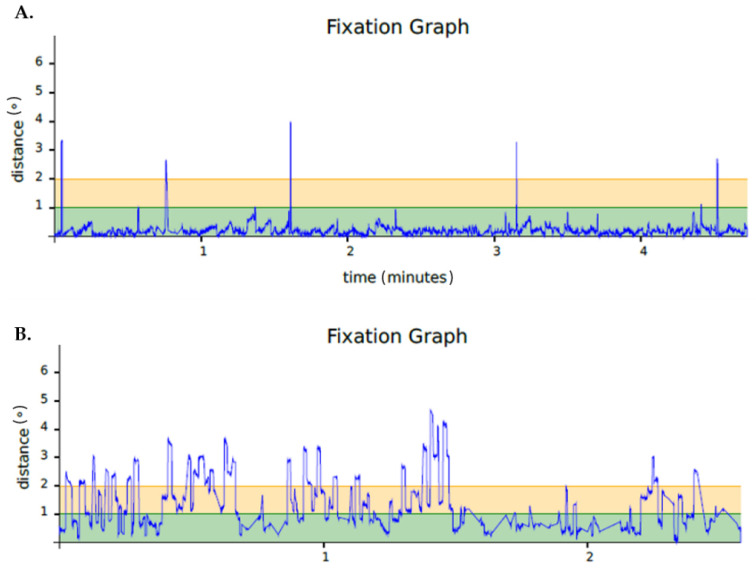
(**A**) Fixation graph of a DLC patient at the beginning of the study. (**B**) Fixation graph of the same patients with follow-up at 12 months.

**Table 1 jpm-11-00698-t001:** Baseline characteristics of the patients.

	T2D NC	T2D MCI	*p* Value
*n* (%)	22 (27.2%)	59 (72.8%)	NA
Age (years)	70.1 ± 3.9	74.1 ± 5.1	0.001
Gender (men) %	13 (59.1%)	38 (64.4%)	0.659
Body mass index (BMI) (kg/m^2^)	30.7 ± 5.2	29.5 ± 4.9	0.323
Smoker (%)	13 (59.1%)	34 (57.6%)	0.988
Hypertension (HTA) (%)	18 (81.8%)	56 (94.9%)	0.062
Dyslipidemia (DLP) (%)	19 (86.3%)	55 (93.2%)	0.329
Obstructive sleep apnea (OSA) (%)	5 (23.8%)	13 (22.8%)	0.926
Ischemic heart disease (%)	5 (22.7%)	19 (32.2%)	0.406
Peripheral arteriopathy (%)	3 (13.6%)	14 (23.7%)	0.321
T2D duration (years)	19.5 ± 8.3	19.2 ± 9.9	0.884
Insulin use	12 (54.5%)	44 (74.6%)	0.083
Insulin dosed (UI/kg/day)	0.59 ± 0.2	0.57 ± 0.3	0.835
Glycosylated hemoglobin (HbA1C DCCT) (%)	7.7 ± 1.1	7.6 ± 1.0	0.665
Severe hypoglycemia at last year (*n*)	0	2	NA
Type 2 diabetes related complications			
Retinopathy (%)	13 (59.1%)	36 (61.1%)	0.875
Nephropathy (%)	8 (36.4%)	25 (42.4%)	0.624
Polyneuropathy (%)	7 (31.8%)	19 (32.2%)	0.974
Mini Mental State Examination (MMSE)	28.9 ± 0.8	25.7 ± 0.9	<0.001

T2D, type 2 diabetes; NC, normo-cognitive; MCI, mild cognitive impairment; BMI, body mass index. NA: not applicable.

**Table 2 jpm-11-00698-t002:** Independent predictors of mild cognitive impairment at baseline.

Coefficients											
Model		Non-Standardized Coefficients		Standardized Coefficients	*t*	Sig.	Correlations			Collinearity	
		B	Standard Error	Beta			Zero	Partial	Part	Tolerance	VIF
1	(Constant)	−0.134	0.862		−0.155	0.877					
	Retinal sensitivity	−0.021	0.010	−0.297	−2.157	0.036	−0.380	−0.300	−0.274	0.853	1.172
	BCEA95	0.022	0.011	0.115	0.809	0.042	0.171	0.117	0.103	0.803	1.245
	Gender	0.074	0.097	0.098	0.760	0.451	0.092	0.110	0.096	0.974	1.027
	Age (years)	0.020	0.011	0.237	1.806	0.077	0.308	0.255	0.229	0.936	1.068
	T2D duration (years)	−0.009	0.006	−0.191	−1.424	0.161	−0.146	−0.203	−0.181	0.893	1.120

Dependent variable: MCI. BCEA95, bivariate contour ellipse area 95.

**Table 3 jpm-11-00698-t003:** Retinal microperimetry parameters at baseline.

	T2D NC	T2D MCI	*p* Value
*n*	22	59	NA
Retinal sensitivity (dB)	23.7 ± 0.3	18.4 ± 0.8	0.0001
Duration (min)	1.9 ± 0.4	2.9 ± 1.0	0.0001
Fixation stability P1 (%)	84.4 ± 4.5	68.9 ± 3.4	0.0124
Fixation stability P2 (%)	91.9 ± 2.6	84.3 ± 3.5	0.0411
BCEA63 (º^2^) (median (range))	0.8 (0.4–2)	2.1 (3.4–18.1)	0.0042
BCEA63 (º^2^) (mean ± 1 standard deviation)	2.1 ± 0.5	3.7 ± 0.6	0.0004
BCEA95 (º^2^)(median (range))	7.5 (3.4–18.1)	18.1 (11.5–36.4)	0.0025
BCEA95 (º^2^)(mean ± 1 standard deviation)	18.7 ± 5.5	30.9 ± 4.8	0.0233

BCEA63, bivariate contour ellipse area 63; BCEA95, bivariate contour ellipse area 95. NC normocognitive, MCI mild cognitive impairment, NA not applicable.

**Table 4 jpm-11-00698-t004:** MMSE and microperimetry parameters at 12 months follow-up.

	Basal	12-Months Follow-Up	*p* Value
**MMSE**			
Normo-cognitive	28.9 ± 0.2	28.4 ± 0.2	0.0774
Mild cognitive impairment	25.7 ± 0.9	24.7 ± 1.4	0.01
**Retinal sensitivity (dB)**			
Normo-cognitive	23.7 ± 0.3	21.8 ± 1.1	0.0759
Mild cognitive impairment	18.4 ± 0.8	17.5 ± 0.8	0.2753
**Fixation stability P1 (%)**			
Normo-cognitive	84.4 ± 4.5	64.6 ± 7.8	0.0204
Mild cognitive impairment	68.3 ± 3.4	58.1 ± 3.9	0.0045
**Fixation stability P2 (%)**			
Normo-cognitive	91.9 ± 2.6	83.3 ± 4.7	0.0342
Mild cognitive impairment	84.2 ± 3.5	77.1 ± 3.1	0.0040
**BCEA63 (º^2^)** (mean ± 1 standard deviation)			
Normo-cognitive	2.1 ± 0.5	3.9 ± 1.5	0.005
Mild cognitive impairment	3.7 ± 0.6	4.9 ± 0.7	0.0287
**BCEA63 (º^2^)** (median (range))			
Normo-cognitive	0.8 (0.4–2)	1.2 (0.4–2.7)	0.0127
Mild cognitive impairment	2.1 (3.4–18.1)	2.3 (3.8–15.2)	0.0434
**BCEA95 (º^2^)** (mean ± 1 standard deviation)			
Normo-cognitive	18.7 ± 5.5	35.4 ± 10.3	0.0075
Mild cognitive impairment	30.9 ± 4.8	41.4 ± 6.2	0.0402
**BCEA95 (º^2^)** (median (range))			
Normo-cognitive	7.5 (3.4–18.1)	9.6 (0.9–24.9)	0.0234
Mild cognitive impairment	18.1 (11.5–36.4)	20.6 (16.7–67.2)	0.0381

BCEA63, bivariate contour ellipse area 63; BCEA95, bivariate contour ellipse area 95; MMSE, mini-mental state evaluation.

## Data Availability

Data will be available on request.

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
