# Peer review of "The Gaze Fixation Assessed by Microperimetry: A Useful Tool for the Monitoring of the Cognitive Function in Patients with Type 2 Diabetes"

_jpm, 2021, doi:10.3390/jpm11080698_

Round 1

Reviewer 1 Report

Thank you for allowing me to review this interesting manuscript. The manuscript is well-written and easy to read and understand.   The authors assess the usefulness of retinal microperimetry as a monitoring tool to evaluate cognitive deterioration. The title and abstract cover the main aspects of the work.   I have minor corrections and suggestions:   1. The introduction provides background and information relevant to the study. Only the abbreviations should be added: • The abbreviation should be written in full: AD - Alzheimer's Disease; • The abbreviation should be written in full: ADA - American Diabetes Association; • The abbreviation should be written in full: MRI - Magnetic Resonance Imaging; • The abbreviation should be written in full: PET - Positron Emission Tomography;   2. The materials and methods, as well the statistical analysis were described clearly and objectively. Only the following abbreviation should be added: • The abbreviation should be written in full: NIA-AA (National Institute on Aging and Alzheimer’s Association).   3. Results: • When the Authors present the results: “From the total of 100 patients with T2D that were evaluated using RBANS, 31 were NC, 64 MCI and 5 had dementia”, it is not clear if these stated results are the results at 12 months follow-up. Please clarify this paragraph to make it clearer. • The same away, please clarify the timepoint of the evaluation stated in the following phrase: “From all the patients, 59 classified as MCI and 22 classified as controls agreed to be included in the study”.    4. The evidence is supported by the discussion, which is a broader theoretical and practical implications of the findings. No further comments concerning the discussion point.   5. Concerning tables and figures:  • The tables within the manuscript are adequate and provide a good understanding of the whole experience. • One suggestion that could be interesting to provide a reader a break from the text, would be to include a picture of a baseline and a 12-month follow-up of one or two patients.   Thank you. Regards,

Author Response

Thank you very much for reviewing our work and we have incorporated your suggestions that considerably increased the value of our work. Please find here our answers to your comments:

 “Thank you for allowing me to review this interesting manuscript. The manuscript is well-written and easy to read and understand.   The authors assess the usefulness of retinal microperimetry as a monitoring tool to evaluate cognitive deterioration. The title and abstract cover the main aspects of the work.   I have minor corrections and suggestions”:  

  1. The introductionprovides background and information relevant to the study. Only the abbreviations should be added: • The abbreviation should be written in full: AD - Alzheimer's Disease; • The abbreviation should be written in full: ADA - American Diabetes Association; • The abbreviation should be written in full: MRI - Magnetic Resonance Imaging; • The abbreviation should be written in full: PET - Positron Emission Tomography;  

Answer: Many thanks for the revision process and your kind comments on our paper. As recommended all the abbreviation have been written in full.

  1. The materials and methods, as well the statistical analysiswere described clearly and objectively. Only the following abbreviation should be added: • The abbreviation should be written in full: NIA-AA (National Institute on Aging and Alzheimer’s Association).  

Answer: Done. Thank you!

  1. Results: • When the Authors present the results: “From the total of 100 patients with T2D that were evaluated using RBANS, 31 were NC, 64 MCI and 5 had dementia”, it is not clear if these stated results are the results at 12 months follow-up. Please clarify this paragraph to make it clearer. • The same away, please clarify the timepoint of the evaluation stated in the following phrase: “From all the patients, 59 classified as MCI and 22 classified as controls agreed to be included in the study”.   

Answer: As recommended this point has been clarified in the revised manuscript (page 4, lines 148-151).

  1. The evidence is supported by the discussion, which is a broader theoretical and practical implications of the findings. No further comments concerning the discussion point.  

Answer: OK. Thank you!

  1. Concerning tables and figures:  • The tables within the manuscript are adequate and provide a good understanding of the whole experience. • One suggestion that could be interesting to provide a reader a break from the text, would be to include a picture of a baseline and a 12-month follow-up of one or two patients.   Thank you. Regards,

Answer:  Following the reviewer’s suggestion, a new figure (Figure 1) showing the evolution of fixation graph in a representative patient in whom a worsening of MMSE was detected has been added to the revised manuscript.  

Yours sincerely, 

Andreea Ciudin and Rafael Simó, corresponding authors

Reviewer 2 Report

This manuscript by Ortiz Zuniga et al. reports clinically relevant and interesting observations on retinal fixation stability during microperimetry and cognitive function in T2D older than 65 years. In addition the authors report significant changes of fixation parameters over a follow-up of 12 months, which were contrasting to the results of CN probands. This prospective study was obviously carefully planned and carried out. Mainly the manuscript is clear and well written. Unfortunately, I see some confusing numeric differences in the results that should either be explained or corrected.

Some details:

Microperimetry is not a direct test of cognitive function. It is a correlation to cognitive function. Therefore the title should be changed. Since gaze fixation was the most informative parameter this should already be expressed in the title. Otherwise readers may easily expect differences in retinal sensitivity.

Page 1 – Line 20: “Patients with T2D>65 years” may be misunderstood as patients with T2D that were suffering from diabetes for more than 65 years. Same applies to line 62 and line 195.

AD appearing in the first sentence of the introduction should be explained. Please check for the correct declaration of all abbreviations. It seems also to apply to ADA in line 50 and to NIA-AA in line 87.

Line 102-103: How were the 22 control patients recruited?

Line 160: “and 12 months follow-up” must be deleted, since Table 3 shows only the baseline results, as far as I can recognize.

The numeric baseline results in Table 3 and the baseline results in Table 4 are not identical. It seems that there have been changes caused by rounding but others cannot be explained by this. Was the number of patients different in the 12 months follow-up? Did not find an explanation to this. Please check the numbers and if necessary correct them.

Author Response

Dear Reviewer, we would likw to thank you very much for reviewing our work and for your valuable comments that have considerably increased the value of our work. We have also changed the title of the manuscript as you suggested, and is more appropiate. Thank you very much! Please find here our answers to your comments:

This manuscript by Ortiz Zuniga et al. reports clinically relevant and interesting observations on retinal fixation stability during microperimetry and cognitive function in T2D older than 65 years. In addition, the authors report significant changes of fixation parameters over a follow-up of 12 months, which were contrasting to the results of CN probands. This prospective study was obviously carefully planned and carried out. Mainly the manuscript is clear and well written. Unfortunately, I see some confusing numeric differences in the results that should either be explained or corrected. Some details:

  1. Microperimetry is not a direct test of cognitive function. It is a correlation to cognitive function. Therefore, the title should be changed. Since gaze fixation was the most informative parameter this should already be expressed in the title. Otherwise readers may easily expect differences in retinal sensitivity.

Answer: We have change the title as recommended, into “The Gaze Fixation  Assessed by Microperimetry: A Useful Tool for the Monitoring of the Cognitive Function in Patients with Type 2 Diabetes”

  1. Page 1 – Line 20: “Patients with T2D>65 years” may be misunderstood as patients with T2D that were suffering from diabetes for more than 65 years. Same applies to line 62 and line 195. AD appearing in the first sentence of the introduction should be explained. Please check for the correct declaration of all abbreviations. It seems also to apply to ADA in line 50 and to NIA-AA in line 87.

Answer: The points raised by the referee have been addressed as recommended.

  1. Line 102-103: How were the 22 control patients recruited?

Answer: Control patients were those patients classified as normocognitive. In order to further clarify this issue the first paragraph of the results has been replaced by the following: “At baseline, from the total of 100 patients with T2D that were evaluated using RBANS, 31 were catalogued as NC, 64 as MCI and 5 had dementia. Of these patients, 59 MCI and 22 NC agreed to continue in the study during the 12-month follow-up”.

  1. Line 160: “and 12 months follow-up” must be deleted, since Table 3 shows only the baseline results, as far as I can recognize.

Answer: Thank you for identifying this error. This has been corrected in the revised manuscript.

  1. The numeric baseline results in Table 3 and the baseline results in Table 4 are not identical. It seems that there have been changes caused by rounding but others cannot be explained by this. Was the number of patients different in the 12 months follow-up? Did not find an explanation to this. Please check the numbers and if necessary correct them.

Answer: Thanks for yours carefully review. The correct numbers are those found in table 4 and the misprints were in table 3. Consequently, we have proceeded to the correction. We apologize for the mistake.

Yours sincerely,

Andreea Ciudin and Rafael Simó, corresponding authors

Round 2

Reviewer 2 Report

The revised version of the manuscript includes all suggested changes. By this the submission could considerably be improved.

Second look some minor deficits were discovered. In table 3 the unit of N is obviously not (%). The units for BCEA63(92) and other data should be defined or explained. Giving (median) or (mean) is not a substitute for the units. Beyond this, would have to be (median [range]) and (mean +/- 1 standard deviation). Further, the number of declared decimals is inconsistent in Table 3 and 4 (one or two). Given the small number of individuals and usual precision of applied measuring method it is appropriate not to give more than 1 decimal for all reproduced data. As in Table 3 the units should be mentioned for all reported numeric results.

Author Response

"The revised version of the manuscript includes all suggested changes. By this the submission could considerably be improved. Second look some minor deficits were discovered. In table 3 the unit of N is obviously not (%). The units for BCEA63(92) and other data should be defined or explained. Giving (median) or (mean) is not a substitute for the units. Beyond this, would have to be (median [range]) and (mean +/- 1 standard deviation). Further, the number of declared decimals is inconsistent in Table 3 and 4 (one or two). Given the small number of individuals and usual precision of applied measuring method it is appropriate not to give more than 1 decimal for all reproduced data. As in Table 3 the units should be mentioned for all reported numeric results."

Answer: Thank you very much for this constructive and careful review of the manuscript.  We incorporated in the text all the changes that were requested. BCEA is a bivariate ellipse are, and is measured in squared degrees (“º2”)- we added it to the Material and Methods - lines 123-124.  Additionally, we completed Table 3 and Table 4 by adding the median [range]) and (mean +/- 1 standard deviation for the BCEA63 and BCEA95 values, as well as the units for all the parameters.

Your sincerely,

Andreea Ciudin and Rafael Simó